# Introducing External Knowledge to Answer Questions with Implicit Temporal Constraints over Knowledge Base

**Wenqing Wu, Zhenfang Zhu \*, Qiang Lu, Dianyuan Zhang and Qiangqiang Guo**

School of Information Science and Electrical Engineering, Shandong Jiao tong University, Jinan 250357, China; winchywwq@163.com (W.W.); lqsdjtu@163.com (Q.L.); csdzdy@163.com (D.Z.); gqqq777@163.com (Q.G.)

\* Correspondence: zhuzf@sdjtu.edu.cn; Tel.: +86-6137-9310-0702

**Abstract:** Knowledge base question answering (KBQA) aims to analyze the semantics of natural language questions and return accurate answers from the knowledge base (KB). More and more studies have applied knowledge bases to question answering systems, and when using a KB to answer a natural language question, there are some words that imply the tense (e.g., original and previous) and play a limiting role in questions. However, most existing methods for KBQA cannot model a question with implicit temporal constraints. In this work, we propose a model based on a bidirectional attentive memory network, which obtains the temporal information in the question through attention mechanisms and external knowledge. Specifically, we encode the external knowledge as vectors, and use additive attention between the question and external knowledge to obtain the temporal information, then further enhance the question vector to increase the accuracy. On the WebQuestions benchmark, our method not only performs better with the overall data, but also has excellent performance regarding questions with implicit temporal constraints, which are separate from the overall data. As we use attention mechanisms, our method also offers better interpretability.

**Keywords:** knowledge base question answering; attention mechanism; external knowledge

## 1. Introduction

A knowledge base (KB) [1] stores a lot of information, which is complex and structured; they describe things (or entities) and their relationships. KB offers a more readable knowledge network for a machine and provides a more natural way to obtain abundant underlying knowledge. Freebase [2] is one such knowledge base that describes and organizes more than 3 billion facts in a consistent ontology. In fact, KB is usually represented as triples [3], such as, (subject, relation, object), where the subject and object represent entities, and the relation describes the semantic relations between subject and object. These triples are often referred to as facts and can be used for answering questions. For example, the triple (Donald Trump, President, America) can be used to answer the question "Who is the president of America". KB is increasingly used for building question answering systems [4,5].

Knowledge base question answering (KBQA) aims to analyze the semantics of natural language questions and return accurate answer from the knowledge base. At present, the methods proposed to tackle the KBQA task can be roughly categorized into two groups: semantic parsing (SP) methods and information retrieval (IR) methods. SP-based methods [6] aim to transform natural language problems into logical expressions through semantic analysis, then they are transformed into a query language such as SPARQL to retrieve the knowledge base and obtain the answer [7]. Although many SP-based methods can achieve good results in the limited domain, many important components in these works, such as vocabularies and rulesets in Combinatory categorial grammar (CCG) [8], are written manually.

Traditional semantic parses [9] require labeled training data and are limited to narrow domains with a small number of logical predicates, but manually labeling data is time-consuming and laborious. Recent studies handle these limitations through the construction of hand-crafted rules or features [6,10], schema matching [11], and using weak supervision from external resources [12].

SP-based methods are still based on symbolic logic and lack flexibility. When analyzing question semantics, it will be affected by the semantic differences between symbols. IR-based methods directly retrieve answers from the KB in light of the information conveyed in the questions. These IR-based methods can adapt better to large and complex KBs, as they do not need hand-made rules. In recent years, with the rapid development of deep learning technology, deep learning is used more and more in KBQA. On the basis of IR-based approaches, many embedding-based methods [13,14] have been proposed and have shown promising results. Compared with the traditional KBQA methods based on symbols, the KBQA method based on representation learning [14] is more robust, and it has gradually exceeded the traditional method in effect. These methods adopt various ways to encode questions and KB subgraphs into a common embedding space, then directly match them in that space, finally typically trained in an end-to-end manner.

Although the above methods have shown good results, they are not satisfactory in some specific problems, such as questions with implicit temporal constraints. In order to solve this problem, we introduce external knowledge on the basis of Bidirectional Attentive Memory Networks (BAMnet) [15], called Temporal Attention Networks (TAnet), that captures the implicit temporal information in question. We use a novel bidirectional attentive mechanism to obtain the temporal information in question in the light of external knowledge. In the experiments, we prove that our method not only shows better results in the original datasets, but also in the data with implicit temporal constraints.

We summarize the contributions of this paper as follows: (1) we introduce external knowledge to solve the questions with implicit temporal constraints; (2) due to the attention mechanism, it offers good interpretability; (3) on the WebQuestions benchmark, our method performs better, and, on the questions with implicit temporal constraints, performs excellently.

The rest of this paper is organized as follows. After introducing related works in Section 2, we describe our proposed methods in Section 3, and then we show our experimental results in Section 4. Finally, we summarize our work and future work in Section 5.

## 2. Related Work

Generally, the solutions of KBQA can be divided into IR-based methods and SP-based methods. SP-based methods aim to transform natural language problems into logical expressions through semantic analysis, such as simple $\lambda - DCS$ [16], query graphs [17], or executable queries, such as SPARQL [5]. Then the logical forms are executed by the corresponding technique and find the answers from the knowledge base. More recently, neural sequence-to-sequence models have been applied to semantic parsing with promising results [18,19], these methods eschew the need for extensive feature engineering.

Some studies have focused on approaches based on weak supervision from either external resources [20], schema matching [11], or using hand-crafted rules and features [6]. A series of studies has been explored to generate semantic query graphs from nature language questions, such as searching partial logical forms via an agenda-based strategy [21], exploiting rich syntactic information in nature language questions [22], using coarse alignment between phrases and predicates [23], or pushing down the disambiguation step into the query evaluation stage [24]. Notably, some SP-based methods try to exploit IR-based techniques [25] by computing the similarity between two sequences as features, utilizing a neural network-based answer type prediction model, or training end-to-end neural symbolic machine via REINFORCE [26]. However, most SP-based methods more or less rely on handcrafted rules or features, which limit their flexibility.

The general process of IR-based methods directly retrieves answers from the KB in light of the information conveyed in the questions [4,27]. Their main difference is how to select the correct answers

from the candidate set. Yao and Van Durme [28] used rules to extract question features from the dependency parse of questions and used relations and properties in the retrieved topic graph as knowledge base features. Then, the production of these two kinds of features was fed into a logistic regression model to classify the question's candidate answers into correct/wrong.

In contrast, we do not use rules, dependency parse results, or hand-crafted features for question understanding. Recently, embedding-based methods for KBQA are becoming more and more popular, Bordes et al. [29] first applied an embedding-based approach for KBQA, afterwards Bordes et al. [30] proposed the idea of subgraph embedding, which encodes more information (e.g., answer path and context) about the candidate answers. In a follow-up work [30], memory networks [31] were used to store candidate answers and could be accessed iteratively to mimic multi-hop reasoning. Different from the above methods that mainly use a bag-of-words (BOW) representation to encode questions and KB resources, Dong et al. [32] and Hao et al. [14] applied more advanced network modules (e.g., convolutional neural networks (CNNs) and long short-term memory networks) to encode questions. Das et al. [33] proposed Hybrid methods, which achieve improved results by leveraging additional knowledge sources, such as free text.

With the development of the attention mechanism [34], bidirectional attention was first proposed applied in machine reading comprehension [35,36] and was then then applied to KBQA. Most embedding-based approaches encode questions and answers independently. Hao et al. [15] proposed a cross-attention mechanism to encode questions according to various candidate answer aspects. Chen, Y et al. [17] goes one step further by modeling the bidirectional interactions between questions and a KB. This work not only modeled the interactions between questions and a KB but also introduced external knowledge to handle questions with implicit temporal constraints through an attention mechanism. As these previous works cannot handle questions with implicit temporal constraints without rules, dependency parse results, or hand-crafted features, therefore we focus on capturing the interactions between external knowledge and questions. We use deep learning (attention mechanism) to handle questions with implicit temporal constraints without rules, dependency parse results, or hand-crafted features.

Another line of related work is applying deep learning techniques for the question answering task. Grefenstette et al. [37] proposed a deep architecture to learn a semantic parser from annotated logic forms of questions. Iyyer et al. [38] introduced dependency-tree recursive neural networks for the quiz bowl game, which asked players to answer an entity for a given paragraph. Yu et al. [39] proposed a bigram model based on convolutional neural networks to select answer sentences from text data. The model learned a similarity function between questions and answer sentences. Yih et al. [40] used convolutional neural networks to answer single-relation questions on REVERB [41]. However, the system worked on relation-entity triples instead of more structured knowledge bases. We can utilize richer information (such as entity types) in structured knowledge bases.

Based on these works [42,43], we introduce external knowledge using attention mechanism to acquire temporal information on knowledge and questions, then further enhance the question vector through information to increase the accuracy of the question answer.

## 3. Model

On the basis of BAMnet, we propose a method to solve the implicit temporal constraints in the natural language question. The model is shown in Figure 1.

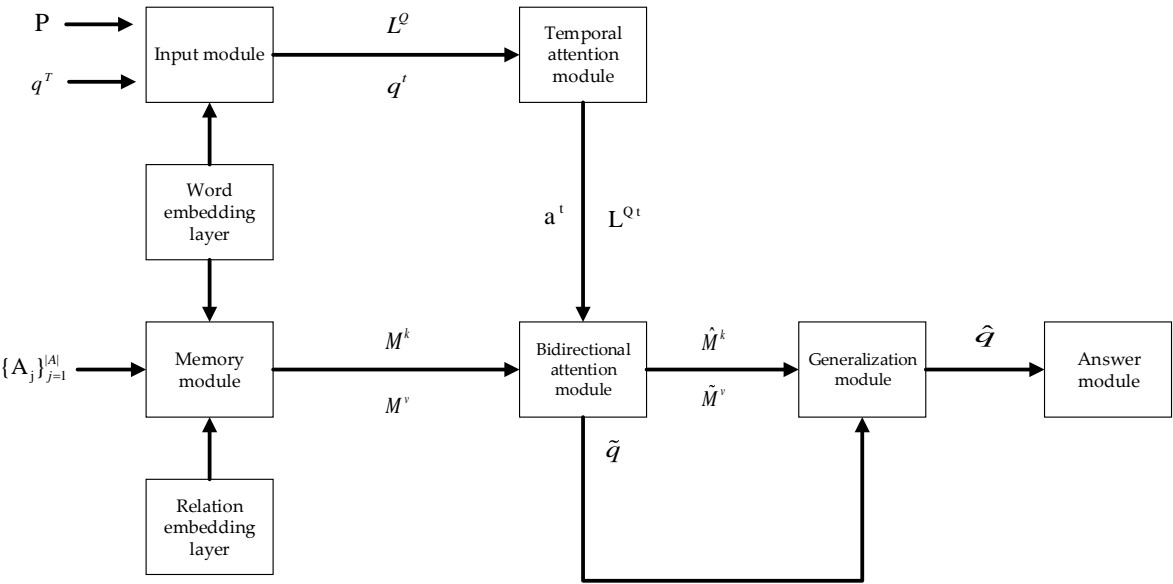

**Figure 1.** Model overview.

### 3.1. Input Module

Formally, an input question $P = p_1, p_2, \ldots, p_m$ was denoted as a word embedding $P^m$ by using a word embedding layer. Then, we encoded the question as $L^Q$ with a bidirectional LSTM [44] (long short-term memory), where $L^Q$ is the sequence of hidden states (i.e., the concatenation of forward and backward hidden states) generated from the BiLSTM.

### 3.2. Memory Module

We used a key-value memory network to store all candidate answers $\left\{ A_j \right\}_{j=1}^{|A|}$ (the closest to main entity and h-hops entity), which were encoded as answer types (entity type in KB), path (sequence of relations from a candidate answer to a topic entity in KB) and context (surrounding entities of a candidate in KB). Using $\left[ M^{kt}; M^{vt} \right]$, $\left[ M^{kp}; M^{vp} \right]$, $\left[ M^{kc}; M^{vc} \right]$ to represent the key-value pair of answer type, path, and context, respectively.

### 3.3. Temporal Attention Module

The temporal information implied in a question is very important for answering the question. In order to solve questions with temporal (e.g., tense) constraints, we proposed a temporal attention module to focus on temporal constraints of a question in Figure 2, which used an attention mechanism to obtain temporal information related to the external knowledge and question, and the external knowledge mentioned earlier are common tense words (such as original, previous, and former). We first used a bidirectional LSTM to encode the pre-processed tense related words $q^T$ as $q^t$:

$$
\begin{aligned}
\overrightarrow{h}_i^t &= LSTM\left( \overrightarrow{h}_{i-1}^t, q^T \right) \\
\overleftarrow{h}_i^t &= LSTM\left( \overleftarrow{h}_{i+1}^t, q^T \right) \\
q^t &= \left[ \overrightarrow{h}_i^t; \overleftarrow{h}_i^t \right]
\end{aligned}
\tag{1}
$$

where the parameters of the LSTM are from those of input layer LSTM and $q^t$, which is fa orward and backward hidden state vector combination of LSTM.

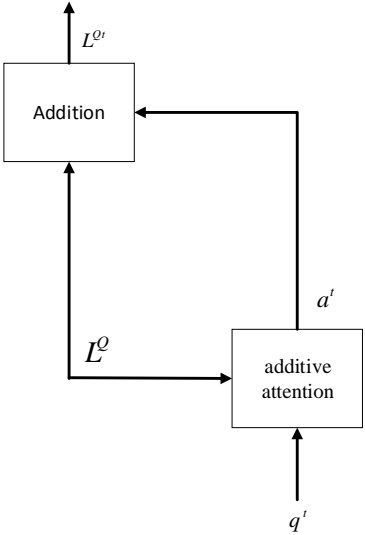

**Figure 2.** Temporal attention module.

Then, we used an additive attention to gain the most relevant temporal information about the question. Putting $q^t$ and $L^Q$ as inputs into the additive attention to obtained implicit temporal information, and in questions $a^t$:

$$a^T = \tanh(W_t L^Q + W_T q^t + b^T)$$
$$a^t = softmax(\sigma(W_t a^T + b^t)) \tag{2}$$

where $\sigma$ is the element-wise sigmoid function; $W_T$ and $W_{T'}$ are the weight matrices corresponding to the question vector $L^Q$ and tense vector $q^t$; $W_t$ is the weight matrix corresponding to their non-linear combination; $b^T$ and $b^t$ are the bias vectors.

We have gained temporal information $a^t$ in view of the question. Then, we integrated the temporal information into the question vector:

$$L^{Qt} = L^Q + a^t. \tag{3}$$

By now, we have obtained temporal-aware question vector $L^{Qt}$, as it contains implicit temporal in light of the question.

*3.4. Bidirectional Attention Module*

The bidirectional attention module aims at catching the connection between the question and knowledge base. As not all components in a question are useful, we focused on the important parts of a question in light of the KB in Figure 3.

We used self-attention for whole new question vector about temporal $L^{Qt}$ to obtain a question vector $q^s$:

$$q^s = BiLSTM\left(\left[L^{Qt}(a^q)^T, L^{Qt}\right]\right.$$
$$a^q = softmax(L^q)$$
$$L^q = \left(L^{Qt}\right)^T L^{Qt} \tag{4}$$

where softmax is used in last dimension of $L^q$. Then, we put $M^{kt}$, $M^{kp}$, $M^{kc}$, and $q^s$ as inputs into additive attention to obtain KB summary:

$$m^y = a^y \cdot M^{vy}$$
$$a^y = softmax\left(\tanh\left(W_1 q^s + W_2 M^{ky}\right)W_3\right) \tag{5}$$

where $y \in \{t, p, c\}$, $W_1$, $W_2$ and $W_3$ are trainable weights, and $M^v$ is the value of the memory network.

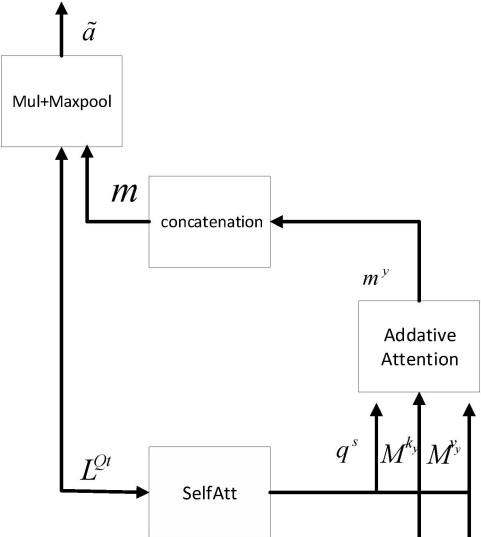

**Figure 3.** The question in light of the knowledge base (KB). Mul: Vector multiplication.

So far, we already have a KB summary in light of the question. Then, we concatenated $m^y$ as $m$. We multiply $L^{Qt}$ and $m$, in order to obtain an attention matrix $A^{QK}$, which is a connection between the question and KB. We used max pooling on $A^{QK}$ to obtain the best connection $a^{QK}$ between the question and KB. Finally, we applied a softmax $a^{QK}$ over to obtain $\widetilde{a}$, which is the importance of the question about the KB.

After finding the vector, which is the question regarding the KB, we continued to obtain a vector that is the KB for the question. First, we obtained a dot-product between $L^{Qt}$ and $M^k$ (concatenation of $M^{kt}$, $M^{kp}$, and $M^{kc}$) to find a connection of the question and the KB attention matrix $A^{qm}$. We used max pooling on the last dimension of $A^{qm}$ and normalized it to obtain the attention matrix $A^x$, which is the importance of the answer aspect for the candidate answer. Then, we continued to compute the question-aware KB representations $\widetilde{M}^k$ and $\widetilde{M}^v$ as follows:

$$
\begin{aligned}
\widetilde{M}^v &= \sum_{i=1}^{3} M_i^v \\
\widetilde{M}^k &= A^x M^k \\
A^x &= softmax\left( \max_j \left\{ A_j^{qm} \right\}_{j=1} \right) \\
A^{qm} &L^{Qt} \left( M^K \right)^T
\end{aligned}
\tag{6}
$$

Then, we enhanced the question and KB representation. We used max pooling on the last dimension of $A^{qm}$ and normalized it to obtain an attention matrix $A^y$, which is the question for the KB attention matrix. Finally, we used $A^y$, the importance of the question about KB $\widetilde{a}$, the KB summary, and $M^v$, the tense vector $a^t$ that we obtained before to find an enhanced question representation $\widetilde{q}$:

$$
\begin{aligned}
\widetilde{q} &= \widetilde{a}\widetilde{L}^{Qt} \\
\widetilde{L}^{Qt} &= L^{Qt} + \widetilde{a} \cdot \left( A^y \widetilde{M}^v \right) + a^t \\
A^y &= softmax(\max_i \left\{ A_i^{qm} \right\}_{i=-1})
\end{aligned}
\tag{7}
$$

Similarly, we enhanced the KB representation $\hat{M}^k$ that included the question information:

$$\hat{M}^k = \widetilde{M}^k + \hat{a}^m \cdot \left( \left( \widetilde{A}^{qm} \right)^T \widetilde{L}^{qt} \right)$$
$$\hat{a}^m = A^y \widetilde{a} \qquad \qquad (8)$$
$$\widetilde{A}^{qm} = softmax(A^{qm})$$

### 3.5. Generalization Module

We used a one-hop attention process before answering. First, we used an attention mechanism to obtain the most relevant information from the memory. Then we renewed the question vector via a GRU (gate recurrent unit) [45]. Finally, we used a residual layer and batch normalization (BN), which can help the model performance in practice. Thus, we have:

$$\hat{q} = BN(\widetilde{q} + \overline{q})$$
$$\overline{q} = GRU(\widetilde{q}, \overline{m})$$
$$\overline{m} = a \cdot \widetilde{M}^v \qquad \qquad (9)$$
$$a = Att_{add}^{GRU}\left( \widetilde{q}, \hat{M}^k \right)$$

### 3.6. Answer Module

Given the question representation $\hat{q}$ and candidate answer representation $\left\{ A_j \right\}_{j=1}^{|A|}$, which is $\left\{ \hat{M}_j^k \right\}_{j=1}^{|A|}$, we computed the matching score $S\left( \hat{q}, \hat{M}_j^k \right)$ between every pair $\left( Q, A_j \right)$ as $S(q,a) = q^T \cdot a$, and ranked their scores to obtain the candidate answers.

## 4. Results

### 4.1. Experimental Datasets

Our experiments were based on the WebQuestions dataset [46], which contains 3,778 training examples and 2,032 test examples. We further split the training data into a training set and validation set, where the training set contained 2298 examples and the validation set contained 755 examples. The validation data is randomly selected from the initial sample. The knowledge base is Freebase KB, which consists of general facts organized as subject–property–object triples. In order to prove the validity of the temporal attention module, we extracted part questions (about 11% of the WebQuestion dataset) and included implicit temporal (e.g., what did James K. Polk do before he was president?) to test what we call t-data. According to some tense related words in the question, we extracted these questions from the test data.

Following Berant et al. [23], macro F1 scores are reported on the WebQuestions test set, where macro F1 scores mean calculate F1 scores through the accuracy and recall rate of each question, and then calculate the average value of F1 scores over all questions. The reason for doing this is that the training and test sets are processed in batches.

### 4.2. Experimental Parameters

When answering the question, not all entities and relations in Freebase will be used, so we only extracted entities and relations that existed in the dataset. The vocabulary size of words is v = 100, 797. There are 1712 entity types and 4996 relation types in the dataset. In particular, the entity may have multiple representations in Freebase, so we only used the entity that is in question. If the entity is boolean values or numbers, we used "bool" and "nums" as their types.

During the training time, we extracted a 2-hop entity, which is close to the topic entity as candidate answers. The memory network size is $M_{max}$=96. We used a pre-trained Glove vector to initialize word embedding with a size $w_v$=300. The relation embedding size $r^e$ and hidden size $h$ were 128. The word

embedding layer, question encoder side, tense words encoder and the answer encoder side in dropout rates were 0.3, 0.3, 0.3, and 0.2. The batch size was 32. In the training process, we used the Adam optimizer [47] to train the model. Initially we set the learning rate as 0.001, then reduced ten times if the performance of the model was not improved in the consecutive epoch. We stopped training if there was no promotion for 20 consecutive times on the verification set.

### 4.3. Results and Analysis

#### 4.3.1. Results

We show the main results of different KBQA method in Table 1.

**Table 1.** Results on the WebQuestions test set.

| Methods (Baseline) | Macro F1 |
| --- | --- |
| SP-based | |
| Yavuz et al. [20] | 0.516 |
| Bao et al. [48] | 0.524 |
| Yih et al. [19] | 0.525 |
| IR-based | |
| Hao et al. [15] | 0.429 |
| Xu et al. [4] | 0.471 |
| BAMnet and our method | |
| BAMnet | 0.557 |
| Our method | 0.563 |

Here, the topic entity is known. Compared to the previous KBQA method (SP-based and IR-based), our method achieved better results with an F1 score of 0.563. We can see that our method is superior to previous state-of-the-art IR-based methods and still remains competitive with SP-based methods, with the effectiveness of bidirectional interaction between question and KB.

It is important to note that compared with the state-of-the-art SP-based methods [19,48], after the introduction of external knowledge, the performance of the method is better than that of SP-based methods and beyond BAMnet. We selected the methods that have performed better in recent years for comparison. For example, Yih et al. [17] used a lot of manual rules to deal with questions with constraints and aggregations, and Bao et al. [48] directly added detected constraint nodes to query graphs to deal with questions with constraints. Yavuz et al. [30] and Bao et al. [48] trained their models on external question answering (Q&A) datasets to obtain extra supervision.

For a fairer comparison, we only show their results without training on external datasets. Although our method also introduces external knowledge, our method is to use deep learning to let the model learn autonomously instead of adding artificial rules. Additionally, the knowledge we introduce is simply words related to tenses, not data sets. Our method uses a deep learning method to handle information and has better interpretability for questions with implicit temporal constraints. Compared to IR-based methods, our method has better performance in WebQuestions. By comparison, our method is sequence to sequence, does not use any rules, and compared to the same sequence to sequence BAMnet, our model can better model questions with implicit temporal constraints. Not only does our model perform better in WebQuestions, but also in the t-data. This fully proves that it is effective to acquire the interactive information between questions and external knowledge through an attention mechanism. The results demonstrate that our method is valid.

#### 4.3.2. Ablation Study

To study the effect of temporal attention module, we conducted ablation analysis under a known topic entity. As shown in Table 2, we can see that temporal attention module is essential to the

performance. Not only does it contribute to the overall model performance but also it performs better on the temporal data we split, suggesting that the introduction of external knowledge is valid.

**Table 2.** Ablation results on the WebQuestions test set.

| Model | Macro F1 |
|---|---|
| Full Model | 0.563 |
| w/o Temporal attention module | 0.556 |

### 4.3.3. Qualitative Analysis

We visualized the attention matrix $a^t$ and checked it whether obtains temporal information in questions. Figure 4 reveals the attention heatmap generated from a test question "who is the current Ohio state senator?". We can see that the attention matrix successfully obtained the temporal information (current) from the question, so we can further strengthen the question vector through the extracted information. In other words, in the question "who is the current ohio state senator?", the word "current" implied temporal information. In the attention matrix, value of the word "current" is the highest, thus there is temporal information in the question, so we can strengthen the question vector in this way.

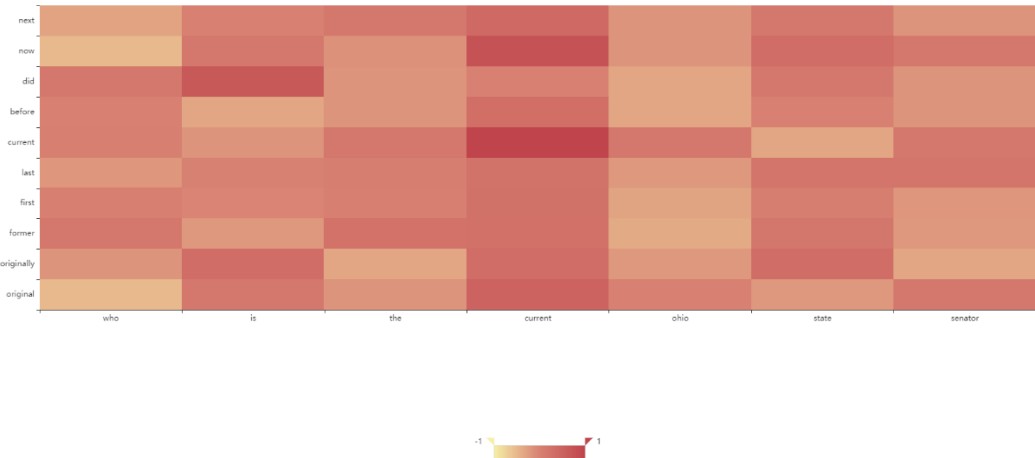

**Figure 4.** Attention heatmap generated by the temporal attention module.

In order to further prove that the introduction of external knowledge is effective, we reveal the predicted answer of our method and BAMnet from the t-data in Table 3. We divided the predicted answers into two categories, which are answer right and answer rank up. Answer right is where the predicted answers are the correct answer but not predicted in other methods, answer rank up is where the predicted answer becomes first place but includes other wrong answers. In the first type, without the temporal attention module, the model cannot capture the information of before, last, and now in the question, so generates the wrong answer, and our method obtains the right answer as the model finds the temporal information through the temporal attention module, it is important that there is no other wrong answer. In the second type, although it includes the correct answer, the model is without the temporal attention module, resulting in the generation of candidate answers, thus, the score of the correct answer is lower than other answers. However, when the model has a temporal attention module, the score of the correct answer is higher than other answers and ranks first. As we can see, compared with other methods without temporal attention module, our method predicts more valid answers and has better performance accuracy.

**Table 3.** Predicted answers of the full model and model w/o the temporal attention module. Where T-att is the temporal attention module.

| Category | Questions | Model w/o T-att | Full Model | Correct Answer |
|---|---|---|---|---|
| answer right | What did James K Polk do before he was president? | Governor of Tennessee, . . . United States Representative | Lawyer | Lawyer |
| | Who did Cliff Lee play for last year? | Cleveland Indians | Philadelphia Phillies | Philadelphia Phillies |
| | Where does Michelle Pfeiffer live now? | Santa Ana | Orange County | Orange County |
| answer rank up | Who was the original voice of Meg Griffin on family guy? | Mila Kunis, Lacey Chabert | Lacey Chabert, Mila Kunis | Lacey Chabert |
| | Where was the first microsoft headquarters located? | Washington, Albuquerque, Redmond | Redmond, Albuquerque | Redmond |
| | Who are the senators of New Jersey now? | Frank Lautenberg, . . . Bob Menendez | Bob Menendez, . . . John Rutherfurd | Bob Menendez |

## 5. Conclusions

In this paper, we present a novel method that obtains temporal information from questions through introducing external knowledge for the purpose of KBQA. Specifically, we encoded external knowledge into the embedding space, obtained temporal information between the question and external knowledge through an attention mechanism, then strengthened the question vector to improve the accuracy. The results show that our method successfully captured the temporal information, and significantly outperformed previous IR-based methods, while remaining competitive with SP-based methods and BAMnet. Qualitative analysis shows that our idea of introducing external knowledge is effective. Although our method works for some questions with implicit temporal constraints, there are some limitations, that is, too many answers will be generated for some complex questions, because complex questions may contain some unknown information. Attention mechanisms have the defect of over learning. In future work, we will explore more effective ways of modeling question with implicit temporal constraints, at the same time, we will address the defects of the attention mechanism to reduce the generation of wrong answers.

**Author Contributions:** Writing—original draft, W.W.; Writing—review and editing, Z.Z.; Data curation, W.W.; Methodology, W.W.; Software, Q.G.; Funding acquisition, W.W.; Project administration, W.W.; Investigation, D.Z.; Validation, Q.L. All authors have read and agreed to the published version of the manuscript.

**Funding:** This work was supported National Social Science Foundation (19BYY076); Science Foundation of Ministry of Education of China(14YJC860042), Shandong Provincial Social Science Planning Project (19BJCJ51, 18CXWJ01, 18BJYJ04).

**Conflicts of Interest:** The authors declare no conflict of interest.

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
