# Peer review of "Introducing External Knowledge to Answer Questions with Implicit Temporal Constraints over Knowledge Base"

_futureinternet, doi:10.3390/fi12030045_

Round 1

Reviewer 1 Report

This study proposes a model based on Bidirectional Attentive Memory Network, which obtains temporal information in question through attention mechanism and external knowledge. Some comments are as follows.

English presentation of this study is poor.  Literature review is not rich. Please update it. More explanation of Fig. 1. Please give more description of experimental datasets Explain what is "Macro F1"

Reviewer 2 Report

Thank you for the submission of your work to this journal. As you have summarized in Section 1, your contributions are solving questions with implicit temporal constraints using attention mechanism on WebQuestion tests.

The question is about your results. Your table shows Macro F1 values. It would be necessary to include the explanation how you obtained these results. Are they directly from the paper or are they something you have obtained using your dataset?

Another relevant to this question is about the validity of the comparison. As it was mentioned in Section 1, the authors have chosen “semi-supervised learning” instead of cleaning the dataset. Maybe, it is not a fair condition for comparison, and it might invalidate the best result out of all listed methods. I may be incorrect but my point is that this article does need the more inclusion of the explanations and justification about the results and comparison. It would be too fast in conclusion about the “superior” to other methods. Authors may need to include solid explanations about their results and comprehensive comparison rather than based on one figure of merit.

In addition, please, refer to the following comments as follows. Thank you for the introducing interesting approach of the deep learning in this knowledge base case.

Line 27: it needs space around the reference [1].

Line 84-86: Not sure of the meaning of this sentence. Please, break the sentence, if needed. Revise this sentence.

Line 91: it needs space between fig. and figure number, and it is recommended to put as figure instead of fig. as it was used in this article.

Figure 1: (1) Font Mismatch and the block in the center needs to be bigger or the text needs to be fit better. (2) Flowchart issue. There is one line drawn on the side. It is not clear about the direction. Does this mean the output of “Temporal attention module” will be feedback to the “Temporal attention module”?

Line 169: Not sure about “we extracted part question” Is this something that needs a grammar correction?

Line 169: “jamesk polk” needs to be “James K. Polk”

Line 173: “question” might have to be “questions” or “the question”

Line 173 – 178: articles (a, s, or the) might be missing

Line 214: Space needed between Fig. and the figure number. And, this must be figure 2 not figure 4.

Table 3. it needs a revision. Current table format is not visually acceptable. Author can adjust the size of columns.

Reviewer 3 Report

Please add a reference to the first sentence int the Introduction section Page 1, line 32: what is triple here? Page 1, line 34: KBQA is not defined Please define all abbreviations before first use The Related work section is rather poor in my opinion, please consider to expand it Why you do not perform cross-validation selection of key parameters (at leas embedding size)? Please explain in the manuscript! Page 7, line 215: you stated: "We can see that attention matrix successfully obtains the temporal information from question, so we can further strengthen question vector through that" - Please explain this claim in great detail. As currently read, it is not self-explanatory! Figure 4 is uninformative in the current settings Conclusion: Please discuss limitations as well as further work opportunities

Round 2

Reviewer 1 Report

I agreed with the revision.

Improve the English presentation as much as possible.

Reviewer 2 Report

This work has been revised. Figure 3 seems to be out of portion. It needs to be resized.

Author Response

Thanks for your suggestions. We have resized the Figure.3.